# Co-Culture of *P*. *gingivalis* and *F*. *nucleatum* Synergistically Elevates IL-6 Expression via TLR4 Signaling in Oral Keratinocytes

**DOI:** 10.3390/ijms25073611

**Published:** 2024-03-23

**Authors:** Lucas Yáñez, Cristopher Soto, Héctor Tapia, Martín Pacheco, Javiera Tapia, Gabriela Osses, Daniela Salinas, Victoria Rojas-Celis, Anilei Hoare, Andrew F. G. Quest, Jessica Díaz-Elizondo, José Manuel Pérez-Donoso, Denisse Bravo

**Affiliations:** 1Microbial Interactions Laboratory, Faculty of Dentistry, Universidad Andrés Bello, Santiago 8370133, Chile; lucas.benjamin.yanez@gmail.com (L.Y.); chrissoto@ug.uchile.cl (C.S.); hector.tapia.g@ug.uchile.cl (H.T.); maartinpacheco@gmail.com (M.P.); javiera.tapia.p@ug.uchile.cl (J.T.); gabriela.osses.s@ug.uchile.cl (G.O.); jessica.diaz.e@unab.cl (J.D.-E.); 2Advanced Center for Chronic Diseases (ACCDiS), Faculty of Medicine, Universidad de Chile, Santiago 8380453, Chile; aquest@u.uchile.cl; 3Oral Microbiology and Immunology Laboratory, Department of Pathology and Oral Medicine, Faculty of Dentistry, Universidad de Chile, Santiago 8380492, Chile; daniela.salinas.diaz@gmail.com (D.S.); a.hoare@odontologia.uchile.cl (A.H.); 4Virology Laboratory, Department of Biology, Faculty of Sciences, Universidad de Chile, Santiago 7800003, Chile; victoria.rojcel@gmail.com; 5Cellular Communication Laboratory, Center for Studies on Exercise, Metabolism and Cancer (CEMC), Program of Cell and Molecular Biology, Institute of Biomedical Sciences (ICBM), Faculty of Medicine, Universidad de Chile, Santiago 8380453, Chile; 6BioNanotechnology and Microbiology Laboratory, Center for Bioinformatics and Integrative Biology (CBIB), Faculty of Life Sciences, Universidad Andrés Bello, Santiago 8370186, Chile; jose.perez@unab.cl

**Keywords:** periodontitis, oral cancer, inflammation, co-culture, *Porphyromonas gingivalis*, *Fusobacterium nucleatum*, toll-like receptor 4, cytokines

## Abstract

Periodontitis, characterized by persistent inflammation in the periodontium, is intricately connected to systemic diseases, including oral cancer. Bacteria, such as *Porphyromonas gingivalis* and *Fusobacterium nucleatum*, play a pivotal role in periodontitis development because they contribute to dysbiosis and tissue destruction. Thus, comprehending the interplay between these bacteria and their impacts on inflammation holds significant relevance in clinical understanding and treatment advancement. In the present work, we explored, for the first time, their impacts on the expressions of pro-inflammatory mediators after infecting oral keratinocytes (OKs) with a co-culture of pre-incubated *P. gingivalis* and *F. nucleatum*. Our results show that the co-culture increases IL-1β, IL-8, and TNF-α expressions, synergistically augments IL-6, and translocates NF-kB to the cell nucleus. These changes in pro-inflammatory mediators—associated with chronic inflammation and cancer—correlate with an increase in cell migration following infection with the co-cultured bacteria or *P. gingivalis* alone. This effect depends on TLR4 because TLR4 knockdown notably impacts IL-6 expression and cell migration. Our study unveils, for the first time, crucial insights into the outcomes of their co-culture on virulence, unraveling the role of bacterial interactions in polymicrobial diseases and potential links to oral cancer.

## 1. Introduction

Periodontitis is a global health issue that affects approximately 10% of the world’s adult population, with even higher prevalence in regions like Latin America and sub-Saharan East Africa [1,2]. Periodontitis is a pathology characterized by chronic inflammation of the periodontium due to a dysbiotic process in the subgingival area of the oral cavity, where the subgingival biofilm resides [3]. This inflammation not only ends up promoting the destruction of the supporting tissues of the teeth but also is related to the etiology of oral cancer, the sixth most common malignancy globally and for which mortality in 2018 reached more than 175,000 deaths [4]. Strikingly, a meta-analysis study indicated that periodontitis could increase the risk for developing oral cancer by nearly twofold [5]. Oral cancer is strongly linked to the compositions of the oral and gut microbiota [6]. In this regard, it has been determined that the saliva microbiota of healthy patients differs from those harboring pre-malignant lesions and oral cancer, reporting increases in certain bacterial phyla, such as *Fusobacterium* and *Bacteroidetes* [7]. Interestingly, among the different bacteria highly associated with oral cancer, *Porphyromonas gingivalis* and *Fusobacterium nucleatum* are considered as two of the most represented bacterial types [6].

*P. gingivalis*, a Gram-negative bacterium found in the oral subgingival biofilm, has been identified as a key player in the pathogenesis of periodontitis [8,9]. It can alter the bacterial community, disrupt the host’s homeostasis, and infect oral epithelial tissue by evading the immune response of the host through various virulence factors [10,11]. Some of these virulence factors include lipopolysaccharide (LPS), which causes cell cycle deregulation; cysteine proteases (gingipains), which destroy the extracellular matrix; and fimbriae, which enable the bacteria to adhere to cell surfaces [12].

Interactions among bacteria within the subgingival biofilm can be physical and metabolic in nature, and these may shape the virulence of specific disease-associated microorganisms [13,14]. *P. gingivalis* has been observed to co-aggregate with other bacteria, such as *Treponema denticola*, leading to a symbiotic relationship. This interaction promotes the exchange of nutrients and increases the expressions of certain *P. gingivalis* virulence factors involved in adhesion, such as hemagglutinin A and gingipains [15].

Furthermore, research in our laboratory has shown that *P. gingivalis* can also interact with *Helicobacter pylori*, also present in the oral bacterial community. This interaction results in increases in the virulence of *P. gingivalis* and the expression of RgpB gingipain and is directly associated with the enhanced migration of infected cells [16].

*P. gingivalis* additionally forms co-aggregates with *Fusobacterium nucleatum*, a filamentous, Gram-negative anaerobe renowned as one of the most prevalent species in the oral microbiome. Moreover, this bacterium exhibits a propensity for binding to numerous periodontal pathogens, a feature pivotal in the advancement of periodontal disease [17,18,19,20]. The interplay between *P. gingivalis* and *F. nucleatum* enhances the ability of the former to survive in oxygen-rich environments [17]. This interaction also leads to alterations in the expressions of *P. gingivalis* proteins, resulting in improved DNA integrity, increased ribosomal content, and consumption of hemin [19]. These changes significantly impact the virulence of *P. gingivalis* [21]. Notably, regarding the interactions of *F. nucleatum* with other oral bacteria, co-incubation studies of *F. nucleatum* and *Prevotella* species have been carried out, reporting enhanced biofilm formation through physical interactions [22].

In studies investigating the co-infection of gingival epithelial cells with *P. gingivalis* and *F. nucleatum*, it was observed that *F. nucleatum* notably heightened the invasiveness of *P. gingivalis* in comparison to infections caused by *P. gingivalis* alone. This co-infection method entailed growing the bacteria as a monoculture, followed by their subsequent mixing at the time of the infection [23]. It is important to note that the approach used in our current work differs as under this condition, the bacteria do not interact before the infection.

Co-culture models are vital for studying interactions between prokaryotic and oral eukaryotic cells, unraveling the pathogenesis of polymicrobial diseases, such as oral conditions [24]. These models enable researchers to explore co-culture dynamics, revealing effects on bacterial gene expression and virulence [15,16,19,25]. In a previous study, it was demonstrated that LPS purified from *P. gingivalis* co-cultured with *F. nucleatum* increased the expressions and secretions of IL-1β, IL-6, and IL-8 in monocytes when compared to the effect of purified LPS derived from monocultured *P. gingivalis* [25]. These results highlight the potential for synergistic interactions between oral bacteria, underscoring the interplay between *P. gingivalis* and *F. nucleatum* and the capacity of the latter to exacerbate the cellular responses induced by *P. gingivalis.*

In terms of in vivo assays, only co-infections with *P. gingivalis* and *F. nucleatum* have been reported. In a mouse model, co-infection with *P. gingivalis* and *F. nucleatum* led to higher releases of IL-1β and TNF-α, as well as an increase in alveolar bone resorption. This effect was likely due to the elevated secretion of these pro-inflammatory cytokines, which are known to be keys in promoting tissue destruction in the oral cavity [26]. Furthermore, in an in vivo periodontitis model with chemically induced tumorigenesis, chronic co-infection of *P. gingivalis* and *F. nucleatum* increased tumor size and invasion capacity via the IL-6/STAT3 axis [27]. Intriguingly, activated STAT3 (phosphorylated STAT3) is associated with oncogenic processes, such as increased cell proliferation, inhibition of apoptosis, and tumor aggressiveness [27]. This suggests a crucial link between these bacteria and the association of periodontitis with oral cancer.

It is important to highlight that IL-1β, IL-6, IL-8, and TNF-α are directly implicated in cancer development/progression [28]. For instance, TNF-α stimulates tumor progression by activating NF-kB and, thereby, regulating processes, like invasion, migration, cell proliferation, and the inhibitions of apoptosis and tumor angiogenesis [29]. IL-6 can initiate tumorigenesis and/or tumor progression by inhibiting dendritic cell differentiation, inducing immune tolerance in the early stages of tumor development, and facilitating metastasis [30]. Furthermore, salivary samples from oral cancer patients have higher IL-8 levels compared to those of healthy individuals or those with pre-malignant lesions [30,31]. Thus, although the roles of these cytokines/pathways in promoting cancer are well documented, the impacts of co-culturing *P. gingivalis* and *F. nucleatum* and, consequently, the interactions occurring in this process, remain to be fully defined in terms of their effects on OK responses.

On the other hand, the molecular mechanisms involved in these responses remain to be defined, whereas the focus of this study was on identifying mechanisms triggering early events that potentially modify epithelial cell behavior to favor the development of cancer later. In this context, several studies have demonstrated that certain bacteria, such as *P. gingivalis*, can enhance the expressions of pro-inflammatory cytokines by activating receptors, like toll-like receptor 4 (TLR4), and, downstream, the transcription factor NF-kB [20,27]. The activation of TLR4 by either *P. gingivalis* or *F. nucleatum* has been associated with the induction of inflammatory responses, including the expressions of cytokines, such as IL-8, IL-1β, IL-6, and TNF-α [16,32,33,34,35,36]. Understanding the mechanisms and implications of TLR4 activation by these bacteria is important for elucidating their roles in periodontal diseases and related systemic conditions.

Based on the aforementioned evidence, our objective was to explore the effects of pre-incubation (co-culturing) and subsequent infection with *P. gingivalis* and *F. nucleatum* on increases in pro-inflammatory mediators associated with early changes in OKs that favor the development of cancer. We also aimed to further our understanding of the mechanisms involved by evaluating the activation of the TLR4 receptor and downstream phosphorylation/translocation of NF-kB to the nucleus.

Shedding light on this link between chronic inflammation and periodontitis will likely aid in improving our understanding of how interactions between periodontitis-associated bacteria may promote the development/progression of oral cancer.

## 2. Results

### 2.1. Standardizing the Culture Conditions and Characterizing the Co-Culture of P. gingivalis and F. nucleatum

Previously, both *P. gingivalis* and *F. nucleatum* had been reported to interact in different ways when co-incubated. Under this condition, *F. nucleatum* promotes greater hemin uptake by *P. gingivalis*, thereby increasing *P. gingivalis*’s virulence [19,21,23]. Nevertheless, there are no reports available regarding the effects of the bacterial co-culture on OKs or any other cell type. Hence, we sought to standardize the co-culture conditions between *P. gingivalis* and *F. nucleatum* in such a way as to achieve a similar ratio between both bacteria. To this end, *P. gingivalis* and *F. nucleatum* were initially grown separately at 37 °C for 24 h until reaching an OD_600_ of ~0.4, and the co-culture was generated as described in the Materials and Methods section.

The ratio of *P. gingivalis* to *F. nucleatum* after 24 h of co-culturing was determined using a colony-forming-unit-per-milliliter counting assay (CFU/mL). Averages of 2.27 × 10^8^ CFU/mL and 3.62 × 10^8^ CFU/mL of *P. gingivalis* and *F. nucleatum*, respectively, were observed, indicating that *P. gingivalis* and *F. nucleatum* were present in a 1:1.59 ratio (Figure 1A). To further confirm the microbial content of the *P. gingivalis* and *F. nucleatum* co-culture, the copy numbers of highly conserved genes were determined using qPCR and primers specific for the 16S rRNA gene of *P. gingivalis* W50 (4 copies) and the *nusG* gene from *F. nucleatum* ATCC 10953 (1 copy) [37,38,39]. No significant differences were observed, indicating that both bacteria were present in similar amounts after 24 h of co-culturing (Figure 1B).

To visualize the presence of both bacteria, co-culture samples were analyzed using scanning electron microscopy (SEM). Aliquots of cultures of *P. gingivalis* W50 and *F. nucleatum* ATCC 10953 (Figure 1C, D, respectively) and the co-culture of both species (Figure 1E,F) were evaluated. Figure 1E, included as an approximate visual representation (×1000 image zoomed in by 61%) of *P. gingivalis* and *F. nucleatum,* reveals that the two bacteria are present in a similar ratio when co-cultured for 24 h. The SEM results suggest that, after 24 h of co-culturing, *P. gingivalis* and *F. nucleatum* might interact physically (see white arrows).

### 2.2. The Co-Culture of P. gingivalis and F. nucleatum Synergistically Increases the Expressions of Pro-Inflammatory Cytokines in OKs

Both *P. gingivalis* and *F. nucleatum* alone promote the expressions and secretions of pro-inflammatory cytokines related to chronic inflammation, such as TNF-α, IL-1β, IL-6, and IL-8, in different cell types; however, there are no reports in this regard concerning OKF6/TERT2 cell cultures [32,40,41,42,43]. Additionally, some reports have described that the co-infection of both bacteria promotes increases in the expressions of TNF-α, IL-1β, and IL-6 in murine in vivo models [26,27]. Thus, OKF6/TERT2 cells were infected with *P. gingivalis*, *F. nucleatum*, the co-culture of both bacteria, and the co-infection of both bacteria and then the mRNA levels of TNF-α, IL-1β, IL-6, and IL-8 were determined using RT-qPCR.

We observed that the mRNA levels of TNF-α increased up to 2.8-fold after infecting OKF6/TERT2 cells with the co-culture of *P. gingivalis* and *F. nucleatum* (Figure 2A). Furthermore, the mRNA levels of IL-1β, IL-8, and IL-6 also increased up to 4.5, 5.8, and 3.7-fold, respectively. Interestingly, for IL-6, these increases occur in a synergistic manner, (Figure 2B–D, respectively). When compared with the non-infected control, no significant changes in any of the cytokines were observed after infecting OKF6/TERT2 cells with either monoculture or co-infecting with *P. gingivalis* and *F. nucleatum*. Taken together, these findings demonstrate that the co-culture of *P. gingivalis* and *F. nucleatum*, rather than the monocultures or the separate infections with the bacteria, increases the expressions of TNF-α, IL-1β, IL-6, and IL-8 in OKs. Intriguingly, the increase in the IL-6 expression is found to be synergistic.

### 2.3. The Co-Culture of P. gingivalis and F. nucleatum Increases NF-kB Phosphorylation in Oral Keratinocytes

The activation of NF-kB is considered as the key step in triggering inflammatory responses associated with several diseases, including periodontitis and cancer [44,45,46]. Both *P. gingivalis* and *F. nucleatum* have been shown to trigger inflammatory responses through either NF-kB or via STAT3 activation in epithelial cells. These responses then augment the expressions of pro-inflammatory cytokines, such as TNF-α and IL-8, as well as metastasis-associated genes [40,42,43,47]. Bearing in mind that the expressions of pro-inflammatory cytokines are linked to the activation(s) of the NF-kB and/or STAT3 signaling pathways [40,42,43,47], we analyzed the changes in the phosphorylation and activation of both transcription factors. Thus, considering that NF-kB and STAT3 are phosphorylated in a time-dependent manner and taking into account that the phosphorylation of both transcription factors peaks after 60–120 min [48,49,50,51], the cells were analyzed using western blotting at 2 h post infection.

The co-culture of *P. gingivalis* and *F. nucleatum* promoted NF-kB phosphorylation at 2 h post infection in OKs compared with the non-infected control (7.3-fold increase) (Figure 3A). On the other hand, no changes in phosphorylated NF-kB levels (pNF-kB) were observed after infecting OKF6/TERT2 cells with either the monocultures of *P. gingivalis* and *F. nucleatum* or following the co-infection with both bacteria grown previously as monocultures. However, it should be noted that no significant increases in STAT3 phosphorylation (pSTAT3) were detected after 2 h of infection under any of the experimental conditions (Figure 3B). Hence, our results demonstrate that infection with the co-culture of *P. gingivalis* and *F. nucleatum* increases NF-kB phosphorylation at 2 h post infection in OKs, but no changes in STAT3 phosphorylation are observed at that timepoint, which correlates with our previous results regarding the expressions of pro-inflammatory cytokines.

The kinetics of NF-kB responses depend on the mode of activation [46,52,53]. Thus, we analyzed pNF-kB levels at 24 h post infection to assess whether this transcription factor was still active at this time. No significant changes were detected in the pNF-kB levels of OKF6/TERT2 cells after 24 h of infection with the co-culture or monocultures of *P. gingivalis* and *F. nucleatum* or after co-infection with these bacteria grown individually (Appendix A). Taken together, our results suggest that the co-culture of *P. gingivalis* and *F. nucleatum* increases NF-kB phosphorylation at 2 h post infection, returning to baseline levels after 24 h. This aligns with previous reports where it was demonstrated that NF-kB activation occurs within 120 min, and at that timepoint, NF-kB would be transcriptionally active, thus promoting the expressions of pro-inflammatory cytokines [48,49,50].

### 2.4. The Co-Culture of P. gingivalis and F. nucleatum Promotes the Nuclear Translocation of NF-kB

NF-kB phosphorylation promotes its translocation to the cell nucleus and the expressions of pro-inflammatory cytokines [52]. Moreover, it has been reported that infection with *P. gingivalis* alone and co-infection with both *P. gingivalis* and *F. nucleatum* grown as monocultures promote NF-kB activation, translocation to the nucleus, and the expressions of pro-inflammatory mediators [20,27,34,54]. Considering the synergic increase in the expressions of pro-inflammatory cytokines and the augmented pNF-kB levels observed after infecting OKs for 2 h with the co-culture of *P. gingivalis* and *F. nucleatum*, we evaluated the translocation of NF-kB to the cell nucleus after 2 h by indirect immunofluorescence analysis.

We observed that the infection of OKF6/TERT2 cells with the co-culture of *P. gingivalis* and *F. nucleatum* significantly increased nuclear NF-kB levels compared with those of the non-infected controls. On the other hand, no changes were detected after infection with the monocultures of *P. gingivalis* and *F. nucleatum* or co-infection with both bacteria previously grown separately. Thus, only the co-culture of *P. gingivalis* and *F. nucleatum* promotes NF-kB translocation to the cell nucleus (Figure 4).

It has been reported that in an in vivo model of chemically induced carcinogenesis, the chronic co-infection of *P. gingivalis* and *F. nucleatum* activated the IL-6/STAT3 axis in the tongue epithelium, which in turn led to augmented tumor size and invasiveness [27]. Taking the above into account and considering our results showing a synergic increase in IL-6 expression after infection with the co-culture of *P. gingivalis* and *F. nucleatum*, we evaluated using indirect immunofluorescence, STAT3 translocation to the cell nucleus at 2 h post infection. As shown, we did not detect significant changes in STAT3 nuclear translocation in OKF6/TERT2 cells under any of the studied conditions (Figure 5), which suggests that STAT3 does not participate in the effects observed after infection with the co-culture of *P. gingivalis* and *F. nucleatum* at that timepoint.

### 2.5. The Co-Culture of P. gingivalis and F. nucleatum Increases the Migration of Infected OKs

Considering our results regarding the expressions of TNF-α, IL-6, and IL-8 (Figure 2), we sought to determine if the co-culture of *P. gingivalis* and *F. nucleatum* promoted the migration and viability of OKs. To do so, we performed MTS (Figure 6A), trypan blue (Figure 6B), and migration assays (Figure 6C). None of the experimental conditions affected the cell viability of OKs regarding the non-infected control, which was assessed using MTS and trypan blue assays. Taken together, these results suggest that neither infection with the monocultures or the co-culture nor co-infection with *P. gingivalis* and *F. nucleatum* affects the viability of OKs under these conditions. In contrast, a significant increase in migration was observed after infecting OKs with the monoculture of *P. gingivalis* (2.4-fold change) and the co-culture of *P. gingivalis* and *F. nucleatum* (3.2-fold change) and the co-infection of both bacteria (4.5-fold change) when compared to the non-infected control (Figure 6C).

### 2.6. The Co-Culture of P. gingivalis and F. nucleatum Does Not Affect TLR4 Levels

Both *P. gingivalis* and *F. nucleatum* have been shown to activate the TLR4 receptor in different cell types, thereby modulating responses, such as cell migration [27,42,43,47,55]. Moreover, NF-kB activation and the expressions of pro-inflammatory mediators are downstream targets of this receptor [42]. Hence, we evaluated whether TLR4 levels correlated with our previous results regarding the expressions of TNF-α, IL-1β, IL-6, IL-8, and NF-kB and cell migration.

Strikingly, neither infections of OKs cultures with monocultures of *P. gingivalis and F. nucleatum* or the co-culture nor co-infection with both bacteria previously grown individually affected the total TLR4 protein levels under the evaluated experimental conditions, which suggests that neither *P. gingivalis* nor *F. nucleatum* alone increases TLR4 levels upon infection (Figure 7). It is important to measure the activation of TLR4 beyond simply determining its expression level. Although measuring the levels of TLR4 provides insight into the abundance of this receptor, assessing its activation status gives a more comprehensive understanding of its functional role in the inflammatory response. Hence, we wondered whether TLR4 was being activated by the co-culture of *P. gingivalis* and *F. nucleatum* in OKs.

### 2.7. The Roles of TLR4 in Pro-Inflammatory Cytokine Expression and Cell Migration

The activation of TLR4 involves a series of downstream signaling events that trigger the production of pro-inflammatory cytokines, chemokines, and other immune mediators. Therefore, evaluating the activation of TLR4 provides valuable information about the extent and nature of the responses initiated upon encountering pathogens, such as *P. gingivalis* and *F. nucleatum* [27,42,43,47,55]. To that end, TLR4 expression was reduced up to 65% using an shRNA approach (refer to shC in our previous work) [16]. To further examine the impacts of the TLR4 knockdown, we analyzed the levels of the total NF-kB in shTLR4 cells that were previously transduced with the shC clone. These cells were infected with the monocultures or co-culture or co-infected with *P. gingivalis* and *F. nucleatum* grown individually. No significant changes were observed in the total NF-kB levels compared with those of the non-infected control under any of the experimental conditions (Appendix A), which demonstrates that the TLR4 knockdown did not affect the total NF-kB levels in OKs.

After infecting shTLR4 cells and subsequently analyzing the mRNA levels of the pro-inflammatory cytokines TNF-α, IL-1β, IL-8, and IL-6 (Figure 8A–D), we observed that infection with the co-culture of *P. gingivalis* and *F. nucleatum* triggered significant increases in the expressions of TNF-α (3.0-fold change) and IL-8 (7.4-fold change) in shTLR4 cells compared with those of the non-infected control. Furthermore, the co-culture also increased the expression of IL-1β (3.4-fold change). Strikingly, the co-culture did not trigger changes in IL-6 mRNA levels (1.5-fold change), demonstrating that there was no increase in the expression of this cytokine in shTLR4 OKs cells (Figure 8D). Infections of shScramble cells were used as control (Appendix A). Taken together, our results suggest that the co-culture of *P. gingivalis* and *F. nucleatum* increases the expression of IL-6 through a TLR4-signaling-dependent pathway. Nonetheless, the precise mechanisms through which this co-culture modulates the expressions of TNF-α, IL-1β, and IL-8 remain to be determined.

In a similar way, we observed no changes in the numbers of migrated cells per field for infected and non-infected shTLR4 cells when compared with the wild-type control cells (WTs). On the other hand, in cells previously transduced with an shScrambled RNA (shScrambled cells), increases in migration were observed following infections with the monoculture of *P. gingivalis* (3.7-fold change) and the co-culture (3.4-fold change) and co-infection with *P. gingivalis* and *F. nucleatum* (2.4-fold change). These findings agree with previous observations for OKF6/TERT2 cells (Figure 6C). Strikingly, we detected that the previous changes in migration observed for shScrambled cells infected with the monoculture of *P. gingivalis* or the co-culture or co-infected with *P. gingivalis* and *F. nucleatum* were not observed in TLR4 knockdown cells. Therefore, our results demonstrate that TLR4 regulates the migration of OKs infected with the monoculture of *P. gingivalis* and the co-culture of *P. gingivalis* and *F. nucleatum*, as well as following co-infection by both bacteria (Figure 8E).

Overall, our results suggest that *P. gingivalis* and *F. nucleatum*, present in similar ratios under our experimental conditions, might interact physically following 24 h of anaerobic co-culturing. Furthermore, we found that the co-culture of *P. gingivalis* and *F. nucleatum* activates NF-kB and triggers its translocation to the cell nucleus. Concordantly, the co-culture of these bacteria triggered significant increases in TNF-α, IL-1β, and IL-8 expressions and a synergistic increase in IL-6 expression in OKs. In agreement with these findings, the co-culture increased the migration of OKs. Importantly, IL-6 expression and OK migration are regulated through the TLR4-signaling pathway. In summary, our results demonstrate, for the first time, the effects of the co-culture of *P. gingivalis* and *F. nucleatum* on OKs and provide insight into the molecular mechanisms underlying these processes.

## 3. Discussion

Microbial interactions play a crucial role in the progression of periodontitis by regulating the expressions of pro-inflammatory mediators, leading to the destruction of the periodontium [3,56]. *P. gingivalis* and *F. nucleatum* are of particular interest in this context as they reside in the subgingival biofilm and co-aggregate, making significant contributions to the development and advancement of periodontitis [13,18,57]. Beyond their impacts on periodontitis, correlations have been established between both *P. gingivalis* and *F. nucleatum* and oral squamous cell carcinoma (OSCC) [27,58,59,60], highlighting the importance for studying the interactions of these bacteria.

There is a lack of studies investigating the effects of infections with the co-culture of *P. gingivalis* and *F. nucleatum* in OKs, which are highly relevant because *P. gingivalis*, when residing in the subgingival biofilm, is present mainly in the top layer of the biofilm, and, therefore, remains in direct contact with the host cells while simultaneously interacting with other bacteria, such as *F. nucleatum* [18,61]. This represents a significant gap in the literature, as evidence suggests that both bacteria can synergistically impact host cells in periodontitis and oral cancer models [25,26,27]. Therefore, we developed a co-culture model of *P. gingivalis* and *F. nucleatum* to evaluate whether the activation of the TLR4 pathway is enhanced, thus augmenting the expressions of pro-inflammatory mediators and cell migration, both processes that are directly related to the progression of periodontitis [16,26,28,30,31] and are linked to the onset of oral cancer [27,29,62,63,64,65]. Because *F. nucleatum* proliferates faster than *P. gingivalis*, we designed the experimental co-culture to achieve similar ratios of both bacteria. This was deemed as necessary to be able to detect the effects of *P. gingivalis* because *F. nucleatum* proliferates significantly more rapidly than *P. gingivalis* and, therefore, tends to obscure possible effects of *P. gingivalis* (Appendix A). Previous reports have lacked details on the growth rate disparity between the two bacteria [23,25], which is crucial for synchronizing their exponential growth phases for infection assays. In fact, our growth curves revealed that *F. nucleatum* reaches its exponential phase faster than *P. gingivalis*. *P. gingivalis* strain W50 has a generation time of ~8.2 h, while *F. nucleatum* strain ATCC 10953 has a shorter 3-h generation time [66,67]. Confirming the co-culture ratio (1:1), CFU and qRT-PCR assays verified that *P. gingivalis* and *F. nucleatum* were present in similar proportions after 24 h of anaerobic co-culturing under the specified conditions, thereby ensuring that the observed effects were not only because of the *F. nucleatum* response.

Intriguingly, the SEM analysis of a co-culture sample revealed the existence of physical interactions between *P. gingivalis* and *F. nucleatum*, which is consistent with previous findings, highlighting the ability of *F. nucleatum* to co-aggregate, particularly with *Bacteroidetes* genus members, such as *P. gingivalis* [57]. These interactions may be facilitated by adhesins of *F. nucleatum*, such as RadD, allowing these bacteria to interact with multiple oral bacteria [18,68]. However, further research is required to identify the specific virulence factors involved in this interaction.

The presence of chronic inflammation in periodontitis has been associated with malignant processes as it is estimated that over 25% of human cancers are caused by chronic inflammation resulting from bacterial or viral infections [65]. Cytokines play a crucial role in regulating diverse cellular processes, like cell proliferation, angiogenesis, migration, and invasion, via specific receptors [29,31,64]. This study focused on measuring the pro-inflammatory cytokines TNF-α, IL-1β, IL-6, and IL-8, pivotal not only in periodontitis but also in cancer-associated chronic inflammation [27,29,31,62,63,64]. These cytokines are strongly linked to periodontal tissue destruction and immune cell recruitment [30,31,65,69]. Moreover, they can promote cancer development through mechanisms such as DNA damage, the regulation of cancer cell migration, and the control of inflammatory responses [65]. Our results demonstrate that TNF-α, IL-6, and IL-8 gene expressions, which have been reported to be overexpressed in OSCC [56], were increased after *P. gingivalis* and *F. nucleatum* co-culture infection, demonstrating an augmented cellular response in OKs. This may be attributable to an increase in bacterial virulence, possibly as a consequence of interactions occurring during co-culturing. However, more studies are required to identify the specific changes in bacterial virulence that occur.

We observed significant increases in TNF-α, IL-1β, and IL-8 expressions and a clear synergistic increase in IL-6 expression with the *P. gingivalis* and *F. nucleatum* co-culture infection (Figure 2B–D). IL-6, implicated in alveolar bone resorption and STAT3 activation via the IL-6/STAT3 axis, enhances malignancy in murine models [27,30]. Beyond its role in periodontitis, IL-6 induces oral carcinogenesis by promoting the hypermethylation of tumor suppressor genes, such as CHFR, GATA5, and PAX6 [70]. The presence of elevated salivary IL-6 in OSCC patients reinforces the notion of a periodontitis–oral cancer link [71]. Thus, IL-6 emerges as a pivotal cytokine connecting periodontitis and oral cancer. Similarly, the co-culture of *P. gingivalis* and *F. nucleatum* synergistically increases IL-8 expression, which is crucial in periodontitis progression because it facilitates gingival epithelial cell migration [72]. The fact that IL-8 is involved in cellular processes, including proliferation, cell cycle regulation, migration, and angiogenesis, underscores its significance not only in periodontitis but also in establishing its connection to the onset of OSCC [73]. Furthermore, IL-8 has been suggested to represent a potential saliva biomarker for OSCC, given its unique elevation observed in OSCC compared with other oral diseases [64,73]. We additionally report a synergistic increase in IL-1β expression after infection with the co-culture of *P. gingivalis* and *F. nucleatum*. This finding is significant as IL-1β, working synergistically with IL-6, promotes alveolar bone resorption, which is a key feature of periodontitis [30]. Both cytokines are implicated in periodontitis development and severity [3]. Our results agree with those in Lee and Baek’s study [25], where the purified LPS of *P. gingivalis* cells previously co-cultured with *F. nucleatum* showed exacerbated virulence, and the stimulation of THP-1 cells increased IL-1β, IL-6, and IL-8 expressions and secretions. Our findings regarding the increased expressions of IL-1β, IL-6, and IL-8 suggest possible increases in virulence in both *P. gingivalis* and *F. nucleatum* when co-cultured for 24 h.

Furthermore, our study revealed a significant 3-fold increase in TNF-α expression with the *P. gingivalis* and *F. nucleatum* co-culture infection. This aligns with prior research indicating that both bacteria separately increase TNF-α expression in various cell types, including OKs [27,29,74]. TNF-α’s involvements in cell migration, alveolar bone resorption, and the expressions of pro-inflammatory molecules, such as IL-1β, IL-6, and IL-8, are well established [16,29,31]. Increased TNF-α levels in samples from OSCC patients and oral, potentially malignant, disorders, along with saliva from OSCC patients, underscore TNF-α’s relevance in oral cancer [71]. The interaction of TNF-α with its receptor promotes malignant cell processes, including migration, invasion, and angiogenesis [16,29,31]. Therefore, our results regarding TNF-α expression support TNF-α’s role in the periodontitis–oral cancer connection, which is crucial for understanding the molecular mechanisms that are involved.

Cell migration is pivotal in periodontitis as it contributes to the formation and progression of periodontal pockets as well as tissue damage and the migration of oral keratinocytes attached to tooth enamel [16]. TNF-α, IL-6, and IL-8 are intimately linked to this process. These cytokines activate NF-kB, regulating genes related to the epithelial–mesenchymal transition (EMT) and fostering an invasive phenotype [29,62,73,75]. In cancer models, TNF-α, IL-6, and IL-8 strongly induce cell migration and in vivo metastasis [29,75,76]. IL-6 and IL-8 are to be highlighted because they promote migration in various epithelial cells and keratinocytes [65,72,77,78]. In cancer, cell migration, which is vital for tumor metastasis, is regulated though chemokines, cytokines, and growth factors, for which the IL-6 pathway is one of the regulators [62]. Similarly, IL-8 contributes to the migration of OKs [72]. In agreement with the results shown in Figure 2, the *P. gingivalis* and *F. nucleatum* co-culture increased OK migration compared to non-infected cells. Intriguingly, migration assays performed on shTLR4 cells revealed that *P. gingivalis* independently regulates cell migration through a TLR4-dependent pathway, irrespective of whether co-culture or co-infection with *F. nucleatum* occurred, which demonstrates that under our experimental conditions, *P. gingivalis* alone could induce cell migration independent of the pro-inflammatory mediators induced by the co-culture.

It has been well documented that NF-kB activation, which is pivotal in periodontitis and cancer-related inflammation, induces the expressions of pro-inflammatory cytokines, including TNF-α, IL-1β, IL-6, and IL-8, in different cell types [43,44,45,46,50,79]. Our results demonstrate increased NF-kB phosphorylation and nuclear translocation in OKs infected with the co-culture, aligning with our previous results regarding TNF-α, IL-1β, IL-6, and IL-8 gene expressions. Because NF-kB activation and the expressions of pro-inflammatory mediators depend on the TLR4 pathway [42] and because both *P. gingivalis* and *F. nucleatum* alone trigger TLR4 activation and the subsequent expressions of pro-inflammatory cytokines in different cell models [27,42,43,47,55], we investigated whether this receptor connected with NF-kB activation occurred in OKs infected with the co-culture of *P. gingivalis* and *F. nucleatum*. The TLR4 total protein levels remained unchanged after infecting OKs with the monocultures or co-culture or co-infection, indicating no impact on TLR4 protein levels. Nevertheless, we demonstrated that the presence of TLR4 is very relevant to this process as the infection of shTLR4 cells with the co-culture failed to induce any increment in IL-6 expression compared to the non-infected control. Additionally, the co-culture increased IL-1β expression up to 3.4-fold, albeit to a lesser extent than the 4.5-fold increase previously observed in non-transduced cells. This suggests that TLR4 plays a major role in regulating IL-6 expression and a partial role in controlling IL-1β expression in OKs infected with the *P. gingivalis* and *F. nucleatum* co-culture. Because TNF-α and IL-8 levels remained unaltered post TLR4 knockdown, other TLR-dependent signaling pathways are likely activated by the co-culture to control their expressions.

In summary, our findings show that the pre-incubation of *P. gingivalis* and *F. nucleatum* and subsequent infection of OKs induce significant increments in the expressions of IL-1β, IL-8, and TNF-α and a synergistic increase in the IL-6 expression compared to the monocultures of *P. gingivalis* and *F. nucleatum* and the co-infection. These results highlight the significance of the impacts of the interactions between bacteria on their respective virulence, surpassing the effects induced by individual bacteria. Furthermore, these cytokines are relevant to both periodontitis and oral cancer promotion, suggesting that these interactions could impact the progressions of both diseases.

In addition, we demonstrated that IL-6 expression is entirely dependent on the presence of TLR4, although its contribution to IL-1β expression is only partial. TLR4 is also necessary for the migrations induced by the co-culture, co-infection, and *P. gingivalis* alone (see proposed model, Figure 9).

This report highlights the importance of co-culture models in shedding light on how bacterial interactions can alter virulence and subsequent cellular responses in the context of polymicrobial diseases. The observations also point toward the potential relevance of the connection between periodontitis and early periodontal events that could modify epithelial cell behavior to favor the development of cancer. Further studies are required to identify specific virulence factors and the underlying mechanisms involved in these processes.

## 4. Materials and Methods

### 4.1. Bacterial Strains and Monoculture Conditions

The reference strains of *P. gingivalis* W50 (ATCC 53978) and *F. nucleatum subsp. polymorphum* (ATCC 10953) were incubated in BHI broth (Oxoid, Basingstoke, UK) supplemented with L-cysteine (0.4 g/L) (Duchefa Biochemie, Harleem, The Netherlands), hemin (Calbiochem, Merck, Darmstadt, Germany) (5 mg/mL), and menadione (Sigma-Aldrich, Burlington, MA, USA) (5 mg/mL). For the cultures in the solid medium, blood agar plates were prepared using 1.5% agar and 5% defibrillated sheep blood (Winkler Ltda., Santiago, Chile), which were supplemented with 1% hemin and 1% menadione. Bacteria were incubated in an anaerobic atmosphere of 90% N_2_, 5% H_2_, and 5% CO_2_ at 37 °C for 3–4 days, which was achieved using both anaerobiosis jars and anaerobiosis generators (AnaeroGen).

To grow *P. gingivalis* and *F. nucleatum* in the liquid medium (broth culture), isolated colonies (inoculum) were selected with a platinum loop from the blood agar plates supplemented as mentioned above. The tubes were incubated for 20–24 h anaerobically at 37 °C until reaching the required optical density, depending on the experiment to be performed.

### 4.2. Bacterial Co-Culture Conditions

To generate a bacterial co-culture of *P. gingivalis* and *F. nucleatum*, planktonic monocultures of both bacteria were incubated at 37 °C for 24 h until reaching an optical density (OD_600_) of 0.4. Then, the co-culture was initiated by mixing 250 µL of the *P. gingivalis* monoculture and 4 µL of the *F. nucleatum* monoculture. In doing so, an approximate ratio of 1:1 between both microorganisms was obtained after 24 h of growth. This was carried out considering that *P. gingivalis* liquid cultures obtained at an OD_600_ of ~0.6 contain 1 × 10^8^ bacteria/mL, and *F. nucleatum* cultures at an OD_600_ of ~0.6 contain 2.5 × 10^10^ bacteria/mL. The co-culture was grown anaerobically at 37 °C in BHI supplemented with 1% hemin, 1% menadione, and 0.4 g/L of cysteine.

### 4.3. Colony-Forming-Unit Assay (CFU/mL)

The bacterial co-culture of *P. gingivalis* and *F. nucleatum* was serially diluted after 24 h of incubation. Each dilution alone (up to 10^6^) was seeded in a blood agar plate (previously supplemented as described above) and incubated for 3–4 days anaerobically at 37 °C. Then, the numbers of colonies formed independently by *P. gingivalis* and *F. nucleatum* were determined.

### 4.4. Cell Line and Culture Conditions

Immortalized human oral keratinocytes OKF6/TERT2 were obtained from Dr. Anna Dongari-Bagtzoglou. The cells were incubated in keratinocyte serum-free medium (KSFM) (Gibco, Thermo Fisher, Waltham, MA, USA) supplemented with 25 µg/mL of bovine pituitary gland (Gibco, Thermo Fisher, Waltham, MA, USA), 0.2 ng/mL of epidermal growth factor (Gibco, Thermo Fisher, Waltham, MA, USA), 0.3 M calcium chloride (Merck, Darmstadt, Germany), and the antibiotics penicillin/streptomycin (Biological Industries, Kibbutz, Israel) and subsequently incubated at 37 °C in an atmosphere of 5% CO_2_. Cells transduced with either a specific shRNA or a control shScrambled RNA were incubated in the presence of 0.5 µg/mL of puromycin (Sigma-Aldrich, Burlington, MA, USA). All the cell cultures were grown to 70–80% confluence as a monolayer before experiments were performed.

### 4.5. TLR4 Knockdown

The knockdown process of the OKF6/TERT2 cell lines using an shRNA approach has been described previously [16]. Briefly, we obtained lentiviral particles containing three clones of the commercial plasmid (shA, shB, and shC) encoding a short hairpin RNA (shRNA) against TLR4 (pLKO.1-puro, purchased from Sigma-Aldrich, Burlington, MA, USA). This plasmid has a puromycin-resistant cassette to avoid plasmid curation. To carry out viral transduction, 3 × 10^5^ OKF6/TERT2 cells were seeded in all the wells of a 6-well plate and incubated for 24 h at 37 °C in an atmosphere of 5% CO_2_. After this, cell cultures were stimulated with the viral suspension for 24 h. As a control, cells were also transduced with a plasmid harboring the coding sequence of an shScrambled RNA in OKF6/TERT2 cells, which was not expected to affect protein expressions.

### 4.6. Infection and Co-Incubation of P. gingivalis and F. nucleatum

After seeding the number of cells required per experiment, the cells were washed and supplemented with KSFM medium in the absence of antibiotics for at least 20 h before carrying out the experiments.

The *P. gingivalis* and *F. nucleatum* inocula were prepared as described before [55]. Bacterial broth cultures were allowed to grow until reaching the exponential phase (OD_600_ ≈ (0.6–0.9)). The volume required to infect the OKs was chosen to generate a multiplicity of infection (MOI) of 100, taking into consideration that—at OD_600_ = 0.6—liquid cultures contain ~1 × 10^8^ and 2.5 × 10^10^ bacteria/mL for *P. gingivalis* and *F. nucleatum*, respectively. Five experimental conditions were compared: a negative control of non-infected cells (NI), monoinfection with *P. gingivalis* (*Pg*), monoinfection with *F. nucleatum* (*Fn*), infection with the co-culture grown for 24 h prior to infection (CC), and co-infection of both microorganisms grown individually and mixed at the time of the infection (CI). All the conditions were chosen to generate a final MOI of 100. Graphical explanations of the co-culture and co-infection are shown in Appendix A.

After 90 min of infection at 37 °C in 5% CO_2_, the cells were washed 3 times with PBS (Gibco, Thermo Fisher, Waltham, MA, USA) and incubated with KSFM medium supplemented with 300 µg/mL of gentamicin and 200 µg/mL of metronidazole for 24 h at 37 °C in 5% CO_2_.

### 4.7. Scanning Electron Microscopy (SEM)

The co-culture of *P. gingivalis* and *F. nucleatum* as well as the monocultures were centrifuged for 5 min at 3354× *g* and room temperature. Then, the supernatant was discarded, and the pellet was resuspended in Milli-Q^®^ distilled water and centrifuged again at 3354× *g* for 5 min at room temperature. This process was repeated twice. Then, the supernatant was discarded, and the obtained whole bacterial pellet was resuspended in 400 µL of 2.5% glutaraldehyde. All the whole-cell samples obtained from the monocultures of *P. gingivalis* and *F. nucleatum* and the co-culture of both bacteria were analyzed using a scanning electron microscope (JEOL, JSM-IT300LV). (JEOL Ltd., Tokyo, Japan).

### 4.8. MTS Cell Viability Assay

In a 96-well plate, 25,000 cells were seeded per experimental condition in technical triplicate. At 24 h post infection, the cell cultures were washed 2–3 times with sterile PBS, and 100 µL of KSFM medium supplemented with antibiotics was then added to each well. Next, each well was incubated with 20 µL of the MTS reagent, as described by the manufacturer (CellTiter 96^®^ Aqueous One Solution cell proliferation assay (MTS)) (Promega, Madison, WI, USA) for 1 h at 37 °C in 5% CO_2_. Finally, the absorbance at 490 nm was evaluated using a plate spectrophotometer.

### 4.9. Western Blot Assay

A total of 2 × 10^6^ OKF6/TERT2 and shTLR4 cells per experimental condition were infected as described above. The cells were washed with cold PBS and then lysed on ice in cold RIPA buffer supplemented with inhibitory protease and phosphatase cocktails according to the manufacturer’s instructions (Sigma-Aldrich, Burlington, MA, USA). Proteins were quantified using the BCA method (Thermo Fisher, Waltham, MA, USA). Afterward, proteins (40 µg) were separated using SDS–PAGE in 10% acrylamide/bisacrylamide gels for each experimental condition. Then, the samples were transferred to a nitrocellulose membrane overnight. Subsequently, the membranes were blocked for 2 h in a solution of 1% bovine serum albumin (BSA) (stock from Thermo Fisher, Waltham, MA, USA) or 5% non-fat milk in Tris–saline (TBS)–Tween^®^ 20 buffer (0.1%) (TBS-T). Then, the membranes were incubated overnight at 4 °C with the following monoclonal primary antibodies: anti-TLR4 (1:2000) (Santa Cruz Biotechnologies, Dallas, TX, USA, SC-293072), anti-NF-kB (1:1000) (Cell Signaling Technology, Danvers, MA, USA, 8242S), anti-pNF-kB (S536) (93H1) (1:1000) (Cell Signaling Technology, Danvers, TX, USA, #3033), anti-STAT3 (1:1000) (Abcam, Cambridge, UK, ab31370), anti-pSTAT3 (phospho Y705) (1:1000) (Abcam, Cambridge, UK, ab76315), and anti-β-actin (1:20,000) (Santa Cruz Biotechnologies, Dallas, TX, USA, SC-47778), all of which were diluted in a 1% solution of BSA/TBS-T. To observe the proteins of interest, the membranes were incubated with anti-mouse (Bio-Rad, Hercules, CA, USA) or anti-rabbit (Santa Cruz Biotechnologies, Dallas, TX, USA, SC-2357) IgG secondary antibodies (both 1:5000) coupled to the enzyme horseradish peroxidase (HRP). Finally, a Clarity Western ECL substrate kit (Bio-Rad, Hercules, CA, USA) was used to detect the primary antibodies bound to the proteins of interest. The images were obtained using Omega Lum^TM^ G hardware, and the bands were quantified using pixel analysis with ImageJ software.

### 4.10. Genomic DNA Purification (gDNA)

The genomic DNA was extracted using a Qiagen Blood & Tissue Kit (Qiagen, Hilden, Germany, 69504) according to the manufacturer’s instructions with slight modifications [38]. Briefly, 10 mL of the culture medium with cells (OD_600_ = (1–1.5)) was spun down at 9183× *g* for 10 min, and the resulting pellet was then resuspended in Tris–EDTA buffer. Subsequently, fresh lysozyme solution (20 mg/mL) was prepared, and the cell suspension was incubated with 130 µL of the lysozyme solution for 30 min at 37 °C. Then, 200 µL of al lysis buffer (Qiagen, Hilden, Germany) and 25 µL of proteinase K (stock at 600 mAU/mL) (Qiagen, Hilden, Germany) were added, and the samples were incubated at 56 °C overnight. The next day, the samples were incubated at 95 °C for 5 min, and DNA was isolated using a commercially available kit according to the instructions of the manufacturer (DNeasy^®^ Blood & Tissue Kit; Qiagen). The purified DNA was eluted with molecular-grade water, and its concentration was measured using a Synergy LX multimode reader (Biotek, Winooski, VT, USA).

### 4.11. Immunofluorescence Assay

A total of 3 × 10^4^ OKF6/TERT2 cells per experimental condition were seeded on 12 mm coverslips and infected as described above, and the coverslips were deposited on a 48-well plate. The cells were then washed with cold PBS and fixed in 4% paraformaldehyde (stock from Thermo Fisher, Waltham, MA, USA) for 30 min at room temperature. Next, the cells were washed and permeabilized for 30 min with 0.5% Triton^®^ X-100 (stock from Merck, Darmstadt, Germany). After this, unspecific binding sites were blocked with a solution of 5% BSA/PBS for 1 h at room temperature. The cover slips were incubated with the following primary antibodies: rabbit monoclonal anti-NF-kB (1:200, Cell Signaling Technology, 8242S) or mouse monoclonal anti-STAT3 (1:200, SC-8019) overnight at 4 °C in a moist, dark chamber. To observe the proteins of interest, blots were incubated with a 1:200 dilution of the secondary antibody conjugated with the fluorescent probe Alexa Fluor^®^ 488 (Jackson ImmunoResearch, West Grove, PA, USA) for 1 h at room temperature in darkness. The cells were then washed with cold PBS and subsequently incubated with PBS–Hoechst (1:10,000) for 10 min to detect the DNA. Finally, the samples on the coverslips were fixed in Dako mounting medium (Faramount aqueous mounting medium, Merck, Darmstadt, Germany) and then viewed under a SP8 confocal microscope. The fluorescence intensity was subsequently calculated using ImageJ software.

### 4.12. Real-Time Quantitative Polymerase Chain Reaction (RT-qPCR)

A total of 1 × 10^6^ OKF6/TERT2 and shTLR4 cells were plated for each experimental condition. The cells were infected under the same experimental conditions as those mentioned above. For the extraction and purification of the mRNA, the TRIzol^®^–chloroform method (Trizol^®^ stock from Invitrogen, Carlsbad, CA, USA) was used, and the extracted genetic material was quantified using a NanoDrop^®^ spectrophotometer (ND-1000 spectrophotometer) at wavelengths of 230 nm, 260 nm, and 280 nm. Next, we proceeded—according to the manufacturer’s instructions—to treat the purified mRNA with the enzyme DNase I (RNase-Free) (New England Biolabs, Woburn, MA, USA). To convert the mRNA to cDNA, a retrotranscription reaction was performed using the M-MLV enzyme (Promega, Madison, WI, USA) according to the manufacturer’s instructions. The reverse transcription PCR reaction was carried out in a thermocycler (Axygen, Union City, CA, USA) according to the following thermal program: 37 °C for 60 min, then 70 °C for 15 min, and, finally, 4 °C for an indefinite time. Subsequently, 500 ng of the cDNA obtained in the previous step was deposited in each well of a 96-well qPCR plate, and the PowerUp ^TM^ SYBR^TM^ Green Master Mix kit (Applied Biosystems, Waltham, MA, USA) was used for amplification. The primers that were used are shown in Table 1.

For the calculation of the obtained relative gene expression, the delta–delta CT (ΔΔCT) method was used, with CT being the threshold value of one cycle. This was conducted by previously calculating the ΔCT value between the genes of interest and the housekeeping genes (β-actin in this case). Then, the ΔCT value of each experimental sample was subtracted from the ΔCT value of the non-infected control, which allowed us to obtain the ΔΔCT value. Each obtained result was squared (2^−ΔΔCT^) to obtain the relative gene expression.

### 4.13. Migration Assays

A total of 3 × 10^5^ OKT6/TERT2 and shTLR4 cells were seeded in a 24-well plate and incubated for 24 h at 37 °C in 5% CO_2_ and then infected as described above. After that, the cells were resuspended in a serum-free medium and added to the top of each Boyden Chamber (Transwell^®^ Costar, 6.5 mm in diameter, 8 μm pore size) previously coated with 2 μg/mL of fibronectin (Sigma-Aldrich, Burlington, MA, USA). Also, a medium with FBS (5%) (stock from Thermo Fisher, Waltham, MA, USA) was added to the bottom chamber to stimulate migration. After 2 h, the inserts were removed and washed gently with PBS, and the cells that migrated to the lower side of the inserts were stained with 0.1% crystal violet in 2% ethanol and counted under an inverted microscope.

### 4.14. Statistical Analysis

All the results were analyzed using either the ANOVA test (one-way ANOVA) or an unpaired *t*-test when appropriate. Dunnett’s post-test and Tukey’s post-test were used to identify statistically significant differences. All the group values were obtained by averaging at least three or more biological replicates. Values of *p* < 0.05 were considered to be statistically significant. All the data were analyzed using PRISM software (version 8.0, GraphPad).

## Figures and Tables

**Figure 1 ijms-25-03611-f001:**
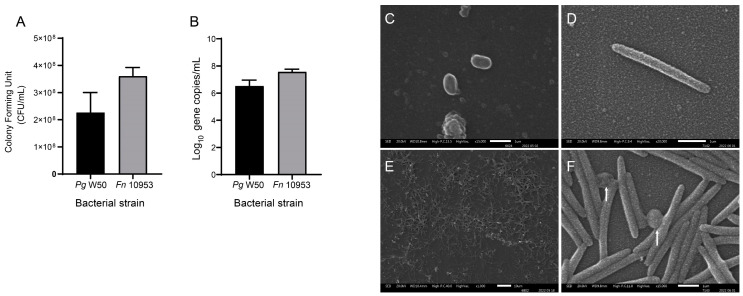
*P. gingivalis* W50 and *F. nucleatum* ATCC 10953 are present in a similar ratio after being co-cultured for 24 h. (**A**) Numbers of colony-forming units per milliliter determined for *P. gingivalis* W50 and *F. nucleatum* ATCC 10953 following the co-culture of both bacteria is shown (*n* = 3). Differences were not significant according to *t*-tests. (**B**) Quantification of copy numbers of 16S rRNA for *P. gingivalis* strain W50 (*Pg* W50) and of *nusG* gene for *F. nucleatum* strain ATCC 10953 (*Fn* 10953) by qPCR following co-culturing for 24 h. No differences were detected; unpaired *t*-test; *n* = 3. (**C**–**F**) Scanning electron microscopy images of aliquots from monocultures of *P. gingivalis* W50 (**C**) and *F. nucleatum* ATCC 10953 (**D**) and the co-culture of both bacteria (**E**,**F**). White arrows suggest physical interactions between *P. gingivalis* and *F. nucleatum*. Scale bar, 1 µm (**C**,**D**,**F**) and 10 µm (**E**).

**Figure 2 ijms-25-03611-f002:**
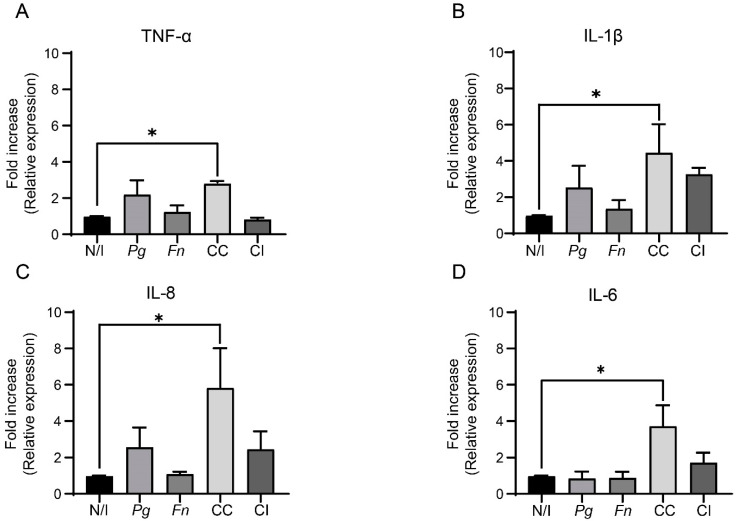
The co-culture of *P. gingivalis* and *F. nucleatum* synergistically increases the expressions of pro-inflammatory cytokines. The fold increases in the relative expressions (ΔΔCTs) of pro-inflammatory cytokines TNF-α (**A**), IL-1β (**B**), IL-8 (**C**), and IL-6 (**D**) in OKF6/TERT2 cells at 24 h post infection. Multiple comparisons were made with the non-infected control. One-way ANOVA; Dunnett post-test. * indicates significant difference of *p* < 0.05; *n* ≥ 4. N/I = non-infected control; *Pg* = *P. gingivalis*; *Fn* = *F. nucleatum*; CC = co-culture; CI = co-infection with both bacteria previously grown separately.

**Figure 3 ijms-25-03611-f003:**
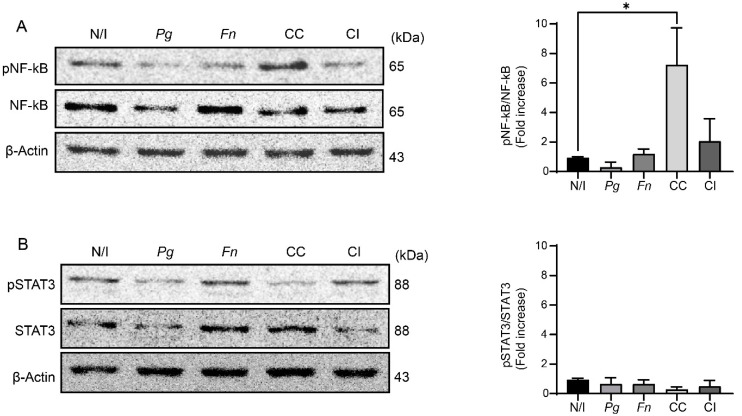
The co-culture of *P. gingivalis* and *F. nucleatum* promotes NF-kB but not STAT3 phosphorylation. Representative images of (**A**) phospho-NF-kB(S536) and total NF-kB and (**B**) phospho-STAT3 (Y705) and total STAT3 for western blot analysis of OKF6/TERT2 cells at 2 h post infection (representative blots to the left), and quantifications by scanning densitometry are shown (graphs to the right). Data were analyzed using one-way ANOVA followed by a Dunnett post-test. * indicates significant difference (*p* < 0.05; *n* = 3). N/I = non-infected control; *Pg* = *P. gingivalis*; *Fn* = *F. nucleatum*; CC = co-culture; CI = co-infection with both bacteria previously grown separately.

**Figure 4 ijms-25-03611-f004:**
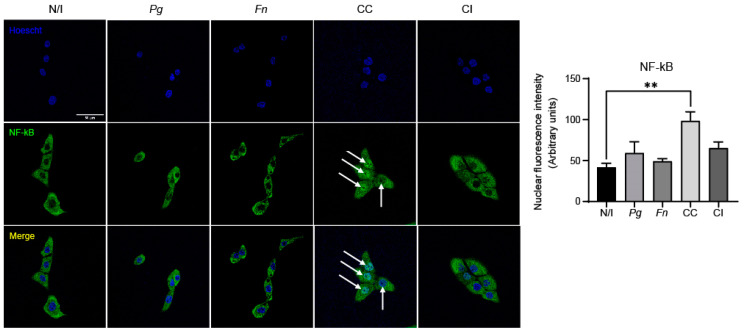
The co-culture of *P. gingivalis* and *F. nucleatum* promotes NF-kB translocation to the cell nucleus at 2 h post infection. Representative images are shown for NF-kB fluorescence (green channel), nuclear staining (blue channel), and the merger of these obtained in OKF6/TERT2 cells at 2 h post infection. Nuclear fluorescence of NF-kB was quantified for all the experimental conditions using imageJ software(version 2.9.0). White arrows indicate nuclear NF-kB. Data were analyzed using one-way ANOVA followed by a Dunnett post-test. ** indicates significant difference (*p* < 0.005; *n* = 3). N/I = non-infected control; *Pg* = *P. gingivalis*; *Fn* = *F. nucleatum*; CC = co-culture; CI = co-infection with both bacteria previously grown separately. Images taken at ×63 magnification. Scale bar, 50 µm.

**Figure 5 ijms-25-03611-f005:**
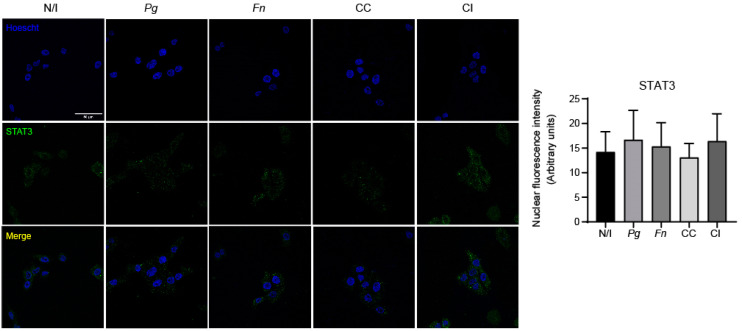
The co-culture of *P. gingivalis* and *F. nucleatum* does not trigger STAT3 nuclear translocation at 2 h post infection in OKs. Representative images of STAT3 fluorescence (green channel), nuclear staining (blue channel), and the merger of these observed in OKF6/TERT2 cells at 2 h post infection are shown. Nuclear fluorescence of STAT3 was quantified for all the experimental conditions using imageJ software(version 2.9.0). Data were analyzed using one-way ANOVA followed by a Dunnett post-test (*n* = 3). N/I = non-infected control; *Pg* = *P. gingivalis*; *Fn* = *F. nucleatum*; CC = co-culture; CI = co-infection with both bacteria previously grown separately. Images taken at ×63 magnification. Scale bar, 50 µm.

**Figure 6 ijms-25-03611-f006:**
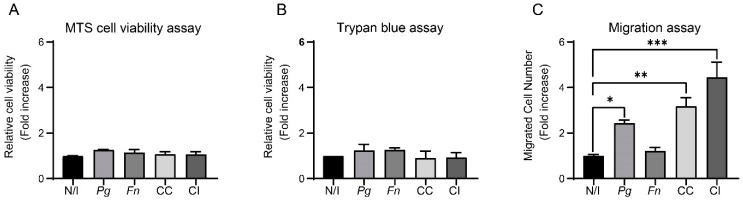
The co-culture of *P. gingivalis* and *F. nucleatum* is not cytotoxic and increases the cell migration of oral keratinocytes. Cell viability of OKF6/TERT2 cells at 24 h post infection was evaluated using MTS (**A**) and trypan blue assays (**B**). Data were analyzed using one-way ANOVA followed by a Dunnett post-test (*n* = 3). (**C**) OKF6/TERT2 cells were infected for 1.5 h, then migration after 2 h was measured. Data correspond to the average number of migrated cells observed in 7 fields. Data were analyzed using one-way ANOVA followed by a Dunnett post-test (*, **, and *** indicate significant differences of *p* < 0.05, *p* < 0.005, and *p* < 0.0005, respectively; *n* = 3). N/I = non-infected control; *Pg* = *P. gingivalis*; *Fn* = *F. nucleatum*; CC = co-culture; CI = co-infection with both bacteria previously grown separately.

**Figure 7 ijms-25-03611-f007:**
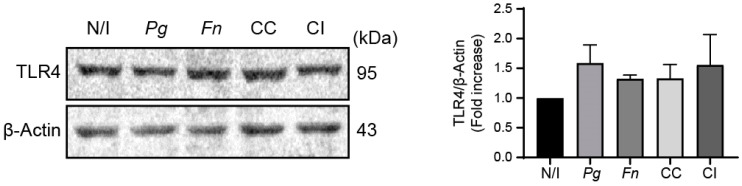
The co-culture of *P. gingivalis* and *F. nucleatum* does not affect TLR4 levels. Representative image of TLR4 levels revealed by western blot analysis of OKF6/TERT2 cells at 24 h post infection (**left panel**), and quantification by scanning densitometry (**right panel**). Data were analyzed using one-way ANOVA followed by a Dunnett post-test (n = 5). N/I = non-infected control; *Pg* = *P. gingivalis*; *Fn* = *F. nucleatum*; CC = co-culture; CI = co-infection with both bacteria previously grown separately.

**Figure 8 ijms-25-03611-f008:**
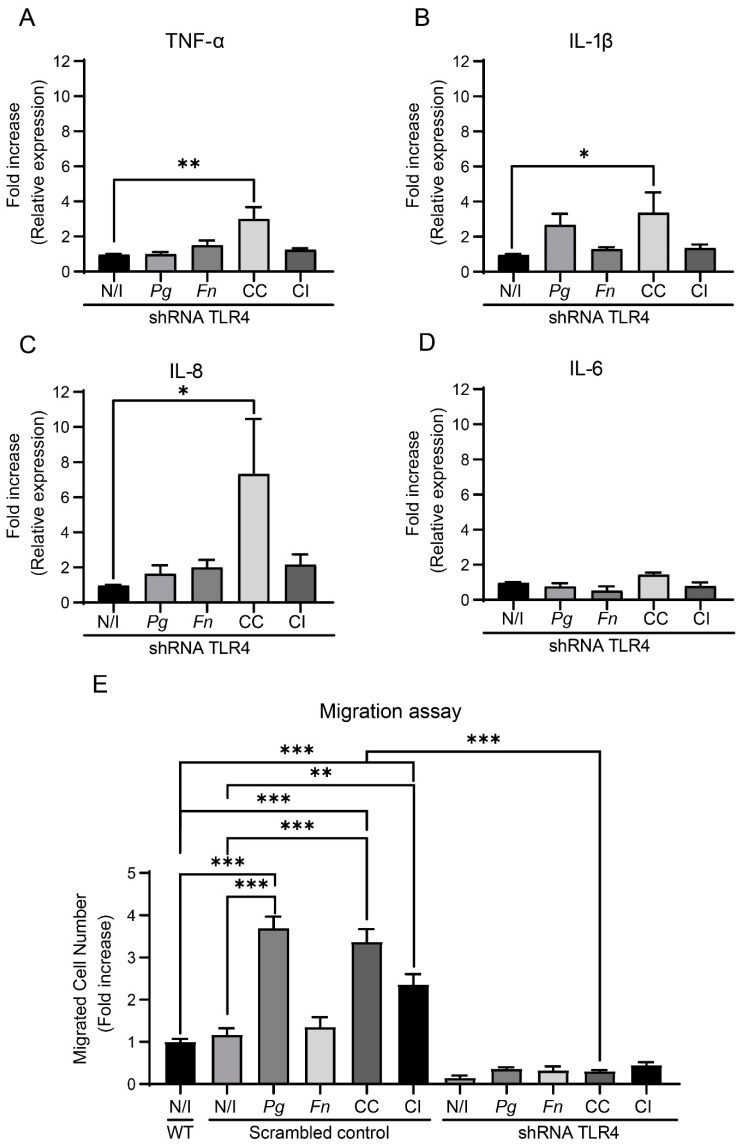
The co-culture of *P. gingivalis* and *F. nucleatum* regulates the expression of IL-6 and cell migration through TLR4 signaling. (**A**) The fold increases in the relative expressions (ΔΔCTs) of pro-inflammatory cytokines TNF-α (**A**), IL-1β (**B**), IL-8 (**C**), and IL-6 (**D**) observed in shTLR4 cells at 24 h post infection are shown. Multiple comparisons were made against the shTLR4 non-infected control. Data were analyzed using one-way ANOVA followed by a Dunnett post-test (* and ** indicate significant differences of *p* < 0.05 and *p* < 0.005, respectively; *n* = 3). (**E**) OKF6/TERT2 cells, shTLR4 cells, and shScrambled cells were infected for 1.5 h, then migration after 2 h was measured. Data correspond to the average numbers of migrated cells observed in 7 fields. Multiple comparisons were carried out using one-way ANOVA followed by Tukey’s post-test (** and *** indicate significant differences of *p* < 0.005 and *p* < 0.0001, respectively; *n* = 3). WT = wild-type control; N/I = non-infected control; *Pg* = *P. gingivalis*; *Fn* = *F. nucleatum*; CC = co-culture; CI = co-infection with both bacteria previously grown separately.

**Figure 9 ijms-25-03611-f009:**
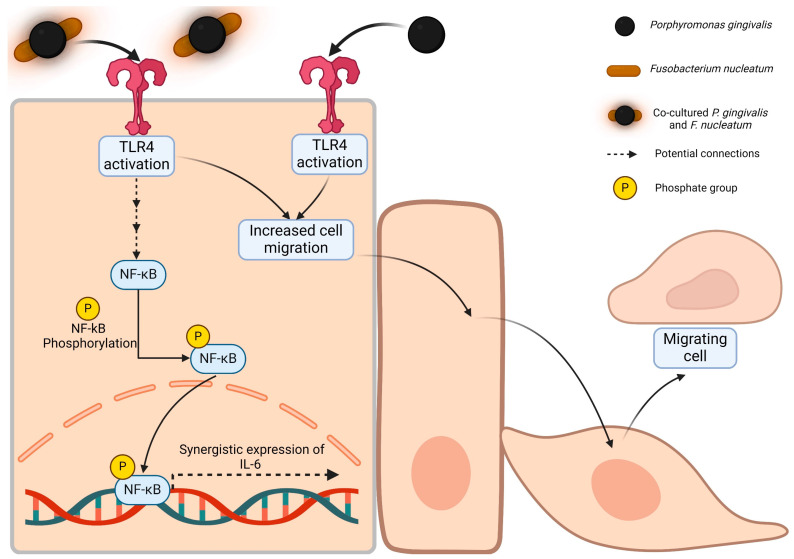
Proposed model of this work. The following model is proposed. For the first time, we demonstrate that the co-culture of *P. gingivalis* and *F. nucleatum* activates the expression of IL-6 and cell migration. We suggest that the pathway involved includes TLR4 because the knockdown of TLR4 blocked this expression. We also observed a correlation between NF-kB phosphorylation and nuclear translocation upon infecting OKs with the bacterial co-culture, suggesting the participation of NF-kB in this process. Moreover, both the co-culture and the monoculture of *P. gingivalis* regulate cell migration through TLR4 signaling (potential connections, as indicated, are illustrated with dashed arrows).

**Table 1 ijms-25-03611-t001:** List of the forward and reverse primer sequences used for each gene of interest.

Gene	Forward Primer (Fw)	Reverse Primer (Rv)
16S rRNA *Pg*	TGTAGATGACTGATGGTGAAAACC	ACGTCATCCCCACCTTCCTC
*Fn nusG* *	CAACCATTACTTTAACTCTACCATGTTCA	TACTGAGGGAGATTATGTAAAAATC
β-actin	CACCATTGGCAATGACGCGTTC	AGGTCTTTGCGGATGTCCACGT
TNF-α	CAGCCTCTTCTCCTTCCTGAT	GCCAGAGGGCTGATTAGAGA
IL-1β	CCACAGACCTTCCAGGAGAATG	GTGCAGTTCAGTGATCGTACAGG
IL-6	AGACAGCCACTCACCTCTTCAG	TTCTGCCAGTGCCTCTTTGCTG
IL-8	GGCACAAACTTTCAGAGACAG	ACACAGAGCTGCAGAAATCAGG

* Gene annotation according to NCBI; Gene ID: 79782952.

## Data Availability

Data are contained within the article and Appendix A.

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
