# Peer review of "Co-Culture of P. gingivalis and F. nucleatum Synergistically Elevates IL-6 Expression via TLR4 Signaling in Oral Keratinocytes"

_ijms, 2024, doi:10.3390/ijms25073611_

Round 1

Reviewer 1 Report

Comments and Suggestions for Authors

This manuscript showed that incubation of co-culture of P. gingivalis and F. nucleatum increase mRNA level of IL-1β, IL-8, and TNFα, and synergistically augments IL-6 expression, and translocate NF-κB to the cell nucleus in gingival epithelial cells (GECs). In addition, they describe that the increase in IL-6 mRNA and cell migration in GECs after incubation with co-culture of these bacteria was dependent on TLR4. Thus, they concluded that interaction of P. gingivalis and F. nucleatum synergistically elevate IL-6 expresssion via TLR4/NF-κB signaling. However, additional experiments are needed to support their conclusion. And, I have comments to improve the manuscript.

1. In title, “Linking Peridontitis and Oral Cancer” is better to remove, because it is not shown in this manuscript.

2. They used co-culture of P. gingvalis and F. nucleatum (around 1:1 ratio). Why did they choose this ratio? Is this ratio physiological condition?  Please explain the reason.

3. In Fig.8A-D, cells after treatment of the scramble shRNA are required as control as shown in Fig. 8E. If not, the effect of tranfection procedure cannnot remove.

4. To support their conclusion, I recommend them to show that knockdown of TLR4 inhibits nuclear traslocation of NF-κB in GECs after incubation with co-culture of P. gingvalis and F. nucleatum.

Author Response

  1. In the title, “Linking Peridontitis and Oral Cancer '' is better to remove, because it is not shown in this manuscript.

R: We agree with the reviewer. As a new title we suggest “Co-culture of P. gingivalis and F. nucleatum synergistically elevate IL-6 expression via TLR4 signaling in oral keratinocytes”.

  1. They used co-culture of P. gingivalis and F. nucleatum (around 1:1 ratio). Why did they choose this ratio? Is this ratio a physiological condition? Please explain the reason.

R: This is an important point that we mention in the discussion section. The physiological ratio is 1:20 for P. gingivalisand F. nucleatum in the subgingival sulcus. However, clinical studies in patients with bleeding reveal that the ratio increases to 1:4 [Abusleme et al, 2013]. For our co-culture experiments, we chose a 1:1 ratio of both bacteria in order to be able to detect effects of P. gingivalis, since F. nucleatum proliferates significantly more rapidly than P. gingivalis (see results in figure S3). This statement was included in discussion section (Page 18 lines 366-367)

  1. In Fig.8A-D, cells after treatment of the scramble shRNA are required as control as shown in Fig. 8E. If not, the effect of transfection procedure cannot remove.

R: As requested, we performed the control experiments suggested, analyzing IL-6 expression levels in shScramble cells. We observed, as expected based on previous results [Soto et al, 2022], similar changes in IL-6 as observed for non-transfected cells (wild type) (see supplementary figure 5).

  1. To support their conclusion, I recommend them to show that knockdown of TLR4 inhibits nuclear translocation of NF-κB in GECs after incubation with co-culture of P. gingivalis and F. nucleatum.

R: We agree with the reviewer that our study does not establish a direct causal link regarding the role of NF-κB in our model. However, in previous studies, it has been established that NF-κB and STAT3 are the key transcription factors involved in producing pro-inflammatory cytokines in OKF6/TERT2 cells [Li et al, 2016, 2019; Zhang et al, 2018]. Our research indicates that NF-κB phosphorylation and its translocation into the cell nucleus were correlated in cells infected with the co-culture, suggesting preferable activation of this pathway rather than STAT3. This observation was described in the discussion section, and we changed: “we investigated whether this receptor regulates the NF-kB activation” to “we investigated whether stimulation of this receptor connects with NF-kB pathway activation (refer to page 22 line 437).

Furthermore, literature supports the idea that the activation of NF-κB and STAT3 is triggered by Toll-like receptors (TLRs) [Kawai and Akira, 2006]. In our model, we confirmed that TLR4 activation is crucial for producing pro-inflammatory cytokines and that the co-culture significantly upregulated IL-6 expression. Considering these findings and the background information provided in the introduction and discussion, it is likely that a connection exists between NF-κB activation (anti-pNF-kB (S536)) and the observed IL-6 expression when infected with the P. gingivalis/F. nucleatum co-cultures. Notably, we have updated our proposed model (refer to Fig. 9), illustrating the connection between TLR4 and NF-κB, as well as NF-κB and IL-6 with dashed lines.

Refer to the attached file for Figures S5 and Figure 9

Reviewer 2 Report

Comments and Suggestions for Authors

This is an interesting in vitro study, which found that co-culture of two important periodontitis pathogens modulates the expression of IL-6 via TLR4/NF-kB pathway in oral epithelial cells. The authors speculated that the results from their research were implicated in the interplay between periodontitis and oral cancer. However, the evidence from this study is indirect and not strong enough to support the authors’ speculation.

Several major and minor comments are listed as follows.

Major comments:

1.     According to the information from Cellosaurus, OKF6 /TERT2 cells are derived from a 57-year-old male patient’s mouth floor. It is not from GINGIVA. However, the current manuscript claims that it is gingival epithelial cells (GECs). The epithelial cells from these two anatomic sites may not be the same.

2.     Since the authors are interested in the linkage between periodontitis and oral cancer, why did they not use representative oral cancer cell lines in some of their important experiments (e.g. migration assay) instead of immortalized oral keratinocytes? There may be significant differences between immortalized oral keratinocytes and oral cancer cells.

Minor comments:

1.     The reagents’ information in M&M seem to be inadequate.

2.     Fig. 1F: The ratio of Pg and Fn seems to be much lower than 1:1. Only two Pg are seen in the photograph while more than 10 Fn are seen.

3.     Fig 3B: The pSTAT3 in CC and Pg groups seems to be down-regulated. The authors did not discuss this result.

4.     Fig 4A: The cell sizes in different groups seem to have discrepancy. They do not look like the same sizes. Why?

5.     Fig 5: The images are not very clear.

6.     References: The formats of the listed references are inconsistent and have some errors. The authors should carefully revise each reference according to the format suggested by the Author Guide.

Comments on the Quality of English Language

Professional English editing is suggested. Some of the English writing is not smooth.

Author Response

Major Comment 1: According to the information from Cellosaurus, OKF6 /TERT2 cells are derived from a 57-year-old male patient’s mouth floor. It is not from GINGIVA. However, the current manuscript claims that it is gingival epithelial cells (GECs). The epithelial cells from these two anatomic sites may not be the same.

R: We agree with the reviewer and we have changed the nomenclature “gingival epithelial cells (GECs)” to “oral keratinocytes (OKs)” throughout the manuscript.

Major comment 2: Since the authors are interested in the linkage between periodontitis and oral cancer, why did they not use representative oral cancer cell lines in some of their important experiments (e.g. migration assay) instead of immortalized oral keratinocytes? There may be significant differences between immortalized oral keratinocytes and oral cancer cells.

R: We agree with the reviewer that cancer cells are a good model to study cancer processes. However, we are interested in studying the early events that could modify epithelial cell behavior to favor the development of cancer later on. Because cancer cells lines are already transformed, they do not represent appropriate models for our experiments. This point is included now in the introduction (Page 3, line 123 and 124; 135 and 136) and discussion sections (Page 23, line 458)

Minor Suggestion 1: The reagents’ information in M&M seem to be inadequate.

R: Thank you for your observation. We have already corrected the information of the reagents in the Materials and Method section according to the journal format. You will be able to see these changes highlighted in yellow (pages 24-28).

 Minor Suggestion 2: Fig. 1F: The ratio of Pg and Fn seems to be much lower than 1:1. Only two Pg are seen in the photograph while more than 10 Fn are seen.

R: The quantification of the ratio in the co-culture is shown in the graphs 1A  and 1B. The image that the reviewer refers to is a magnification (X15,000) of a selected field just to show that direct physical interaction between P. gingivalis and F. nucleatum can occur.

Minor Suggestion 3: Fig 3B: The pSTAT3 in CC and Pg groups seems to be down-regulated. The authors did not discuss this result.

R: The reviewer's observation is something we also noted but after performing three independent experiments in triplicate and quantifying those results, no statistically significant differences were observed (p=0.3372).

Minor Suggestion 4: Fig 4A: The cell sizes in different groups seem to have discrepancy. They do not look like the same sizes. Why?

R: Thanks for your comment. This could be because OKF6/TERT2 cells are non-malignant and therefore exhibit contact inhibition. Consequently, an isolated cell will expand to a greater extent in the absence of adjacent cells.

Minor Suggestion 5: Fig 5: The images are not very clear.

R: Thank you for your comment. The low fluorescence intensity observed may be attributable to the low basal levels of STAT3 in these cells. However, STAT3 levels were also quantified by Western blotting (n=3) and no differences were detected.

Minor Suggestion 6: References: The formats of the listed references are inconsistent and have some errors. The authors should carefully revise each reference according to the format suggested by the Author Guide.

R: We thank the reviewer for this comment. As suggested, this has been corrected.

As requested, English was improved.

Round 2

Reviewer 1 Report

Comments and Suggestions for Authors

Authors almost full my requests in revised manuscript.  I don't have any suggestion anymore.